# Enhancing Network Availability: An Optimization Approach

**Yaser Al Mtawa** 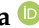

Applied Computer Science, The University of Winnipeg, Winnipeg, MB R3B 2E9, Canada;
y.almtawa@uwinnipeg.ca

**Abstract:** High availability is vital for network operators to ensure reliable services. Network faults can disrupt functionality and require quick recovery. Multipath networking enhances availability through load balancing and optimal link utilization. However, equal-cost multipath (ECMP) routing has limitations in effectively using multipaths, decreasing network availability. This paper proposes a three-phase disjoint-path framework that improves availability by directing traffic flows through separate paths. The framework provides effective load balancing and meets various service requirements. It includes the Optimization phase for identifying optimal multipath solutions, the Path Separation phase for dividing the multipath into working and backup sets, and the Quality Assessment phase for evaluating the robustness of both sets using topological metrics and micro-based characteristics. The simulations demonstrate the proposed framework's validation and effectiveness in enhancing network availability.

**Keywords:** network availability; edge-disjoint multipath; Mixed Integer Linear Programming

## 1. Introduction

A communication network comprises nodes, like routers and switches, connected by links. Routing protocols help share information and avoid congestion as the network grows. Network performance involves Traffic Routing (TR) and Traffic Engineering (TE). TR covers segment routing (SR) and Multiprotocol Label Switching (MPLS)/IP-based networks. SR uses a path with segment IDs, while MPLS/IP routing uses labels given by edge routers. SR relies on sender information in packet headers, but MPLS/IP routing depends on forwarding tables or label data [1].

One of the significant challenges in communication networks is ensuring service availability, as failures can lead to packet loss, retransmission, and disruption. Network availability signifies the operational proportion of a network for optimal performance. It assesses the reliability and effectiveness of computer networks and communication systems. High network availability is vital for seamless communication, data sharing, and workflow management with minimal downtime [2]. TE strives to guarantee Quality of Service (QoS) requirements such as latency, packet loss, throughput, and availability.

Network availability and reachability depend on nodes and links. Problems with these parts can affect the whole network's performance, reliability, and flexibility. To avoid interruptions and issues, it is important to have constantly available nodes and strong links with backup options. Ultimately, maintaining good node and link availability is key for ensuring smooth communication and data transfer in any network [3–5].

Path diversity is a crucial aspect that enhances network availability by offering multiple routes for traffic delivery when rerouting becomes necessary. Disjoint paths can either be link-disjoint or node-disjoint. Path diversity between a pair of nodes, s and *t*, can be calculated as a minimum edge-cut set using Menger's theorem [6]. The Ford–Fulkerson (or Edmonds–Karp) method can be utilized to create multiple distinct paths linking pairs of nodes. However, this approach only offers linearly independent path variety. This means a limited number of entirely separate linear paths are available in a network. Consequently,

bandwidth may be exploited and link-sharing with other traffic could be utilized, increasing the likelihood of network overloading [7].

Despite the distributed nature of routing protocols, the computation of disjoint paths requires a centralized overview. Assume a communication network as the one depicted in Figure 1. If the two nodes $P_1$ and $P_2$ require disjoint paths to reach the same target node, $R_4$, then each of them cannot establish such a path without having full knowledge of the network's link allocation.

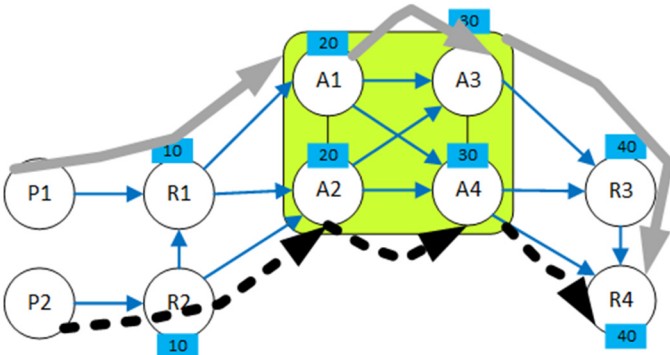

**Figure 1.** An illustration of linear edge-disjoint paths with different source nodes and the same target node with blue boxes indicates the cost/link-metric of the path so far.

On the other hand, Internet interior gateway protocols (IGPs), like Open Shortest Path First (OSPF), navigate based on the shortest routes. When multiple shortest paths exist between a source and a destination, traffic flow is divided equally among all outgoing links belonging to the shortest paths in a process known as the Equal-Cost Multipath (ECMP) principle. Nevertheless, OSPF relies entirely on the weights or costs assigned to network links. Consequently, determining OSPF weights becomes an optimization challenge. Relying solely on link metrics of IGP for weight assignment may lead to network congestion as more traffic flows utilize only OSPF paths [8]. Essentially, IGPs are driven by graph topology, which overlooks bandwidth resources and traffic characteristics when making traffic steering decisions.

The aforementioned method and algorithms primarily offer narrowly separable linear routes or rely on the link weight to identify the shortest paths. Consequently, networks are highly likely to experience overloading and congestion, ultimately leading to unavailability issues.

Considering the challenges above, this paper extends our work in [9] and presents a framework designed to address the limitations of linear independent disjoint paths and the weight-setting prerequisites for OSPF optimization, focusing on critical objective functions such as congestion and utilization. Consequently, network operators may employ all available disjoint paths that align with their network routing optimization goals. Utilizing two sets of disjoint multipaths—working (W) and backup (B)—can significantly assist network operators in managing their traffic reliably. Note that the paths in both W and B sets are not required to be internally disjoint.

The rest of this article is structured in the following way: Section 2 presents related work. Network modeling and problem formulation will be addressed in Section 3. Section 4 is devoted to discussing availability metric in network analysis. Problem statement and our proposed availability framework will be provided in Sections 5 and 6, respectively. Experimental results, which contain the validation and effectiveness of our framework, will be conducted in Section 7. We conclude the paper in Section 8.

## 2. Related Work

Machine learning (ML) models have recently been utilized to identify multipath routing strategies and enhance network performance. Integrating ML techniques with communication networks offers promising prospects for improved network management and

availability [10,11]. Additionally, genetic algorithms have been used to propose a multipath routing protocol which can enhance network security and prolong its lifespan [12,13].

The problem of finding edge-disjoint paths (EDP) is different from calculating the maximum number of EDP between a pair of terminal nodes. Menger's theorem [6] and Edmonds et al. [14] enable finding the maximum number of edge-disjoint paths. On the other hand, Yen's algorithm [15] calculates the single-source K-shortest paths for a graph consisting of nonnegative link costs. This iterative method utilizes any shortest path algorithms, such as Dijkstra's algorithm [16], to determine the optimal or shortest path. It uncovers the following *K-1* deviations from the best path in subsequent iterations. Likewise, Open Shortest Path First (OSPF) identifies the shortest routes between two endpoint nodes. Nevertheless, OSPF implements the Equal-Cost Multipath (ECMP) concept to distribute traffic equally across outgoing links associated with the shortest paths. Neither Yen's algorithm nor OSPF offers edge-disjoint paths, necessitating the optimization of network link weight values to achieve the desired application objective function.

Finding the maximum EDP problem is known to be an NP-hard optimization problem [17]. Martin et al. [17] combined Integer Linear Programming (ILP) and an evolutionary algorithm to deal with the hardness of this problem. Most recently, Pereira et al. [18] proposed an Evolutionary Computation approach to encode routing paths using only three segments. Similarly, Li et al. [19] provided Mixed Integer Linear Programming (MILP) to transmit traffic load between two terminal nodes using paths with no more than K segment to minimize the maximum link utilization in the topology. Unlike [19], which finds multiple paths with traffic splitting, [20,21] focused on finding one K-segment routing for each flow with no traffic splitting. Suurballe [22] introduced an algorithm for finding K-node-disjoint paths with minimal cost, where K is known a priori. The time complexity of this algorithm is $O(kn\log n)$, which is $k$ times the complexity of a single shortest path algorithm. Later research by Suurballe et al. [23] addressed the problem of identifying the shortest pairs of disjoint paths.

It should be noted that the availability framework presented in our study differs from those in [18–21]. While the latter works focus on enhancing network performance by optimizing Segment Identifier (SID) stack size, they aim to maximize the ratio of node SIDs to adjacent SIDs. However, this encoding-based methodology has a disadvantage: it reduces the number of paths in the resulting multipath. Conversely, our framework's goal is to optimize link utilization without constraining the size of the generated multipath. In contrast to the studies presented in references [19–21], which do not consider edge-disjoint multipaths, our research additionally emphasizes network availability.

In this paper, we enable network operators to exploit the features and capabilities of the network routing paradigm, take advantage of the equal cost multipaths (ECMP), and seek load balancing within a network. The key contributions are as follows:

1.  A novel two-stage framework is proposed for addressing the edge-disjoint multipaths problem by formulating a MILP problem that optimizes traffic routing through multipath, aiming to minimize the maximum utilization of network links.
2.  A newly designed splitting algorithm, LDCR, is introduced. While the MILP generates a single multipath, LDCR is responsible for dividing this multipath into two edge-disjoint multipaths: working and backup.
3.  The paper provides a qualitative analysis of the working and backup sets for evaluating the splitting quality of these multipaths by employing edge-cut set, influence indicator, and nodal degree metrics.

Extensive experiments have been conducted on various network topologies to demonstrate the validity and effectiveness of the proposed framework and splitting scheme, using networkwide cost and availability as the primary performance metrics.

## 3. Network Modeling

A communication network can be represented as a graph $G = (N, L)$, where $N$ denotes the set of vertices/nodes that represents the network's elements, and $L$ is the set of

edges/links representing the connectivity between its nodes. $L \subseteq \{\{i, j\}|i, j \in N \text{ and } i \neq j\}$, i.e., undirected graph. Link direction is demonstrated using an ordered pair of nodes: $LD = \{(i, j), (j, i) \mid \{i, j\} \in L\}$. Directed link $(i, j)$ (or $e_{i,j}$) is also called an arc over which data traffic flows from *i to j*. Each arc $e_{i,j}$ has a traffic engineering (TE)-numerical metric value $a_{i,j}$ which is a real number. Let $n = |N|$ and $m = |L|$ be the number of nodes and links, respectively. This paper primarily concentrates on links as elements of *L* that may experience failure. A simple path between two nodes, *s* and *t*, can be described as a collection of adjacent links connecting *s* and *t* without recurring nodes. Multiple paths with varying costs or lengths might exist to connect *s* and *t*. Multipath between $(s, t)$ is defined as a basic representation of a segment/label's core principles and its application within a network. Each multipath arc possesses a bandwidth fraction (BWFrac) and capacity.

Figure 2 demonstrates the multipath along with its associated characteristics, such as BWFrac, displayed in a small gray box above each link. Additionally, the cost of the shortest path from the source nodes to all other nodes is presented in a small blue box above each respective node. In Figure 2, two source nodes, $P_1$ and $P_2$, one target node, $R_4$, and one anycast group A (i.e., anycast is a collection of nodes, A1–A4) are present. Several paths connect $P_1$ and $P_2$ to $R_4$. Examples are the following shortest paths: $P_1 \rightarrow R_1 \rightarrow A_2 \rightarrow A_4 \rightarrow R_4$; $P_2 \rightarrow R_2 \rightarrow A_2 \rightarrow A_4 \rightarrow R_4$. While the cost of these only shortest paths is 40, other paths of this multipath have higher costs.

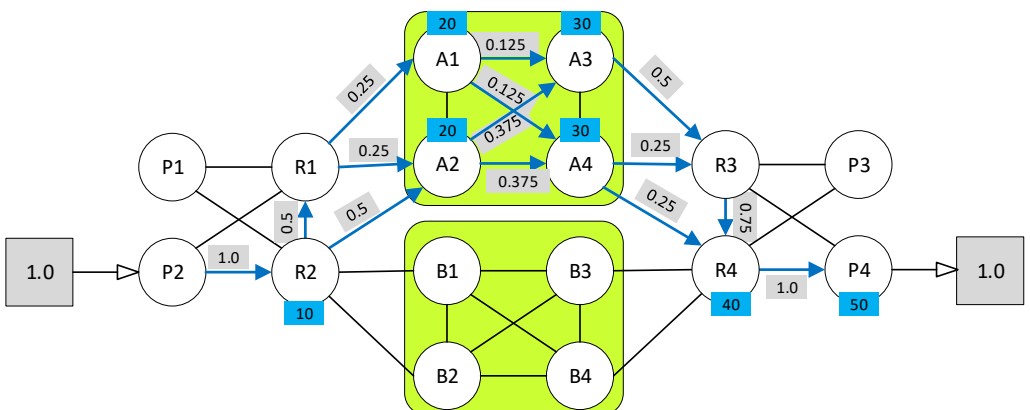

**Figure 2.** An illustration of the concept of multipath using a network with two anycast groups (A1–A4 and B1–B4). Traffic flows with even splitting through ECMPs. Path from nodes P2 to P4 through R2, anycast A1–A4, and R4.

The cost metric of a path is determined by the combined cost metrics of its individual links. In a communication network, each link, $e_{i,j}$, has an associated failure probability, represented as $Pr\left(e_{i,j} \text{ is failed}\right)$. A path, *P*, is operational with probability $Pr\left(P \text{ is operational}\right)$, which depends on the failure probability of its constituent links.

In this paper, we made the following assumptions: (1) The communication network is depicted as a simple undirected graph, with no more than one link between any pair of nodes. (2) While our methodology can be applied to elements of *V*, including hosts, switches, and routers, we focus on allowing only links to fail independently with known probabilities, except those in the same Shared Risk Link Group (SRLG), which fail collectively. (3) Once a link fails or is removed, it remains non-operational and is not restored.

## 4. Availability Metric in Network Analysis

This section delves into the concept of the availability metric and its significance in network analysis. The discussion presented here is broad and independent of specific network topologies. Network availability is crucial for operators and end-users, particularly due to the high demand for real-time services. Failures in network elements, such as links, can cause unreliable services. As a result, network operation availability is heavily impacted

by the degradation of these elements. A dependable network must continue functioning even in the face of increased faults.

In essence, availability refers to the likelihood that a system will continue to deliver packets among its devices. In communication networks, an availability metric measures the network's resilience following potential failures. Consequently, network operators must fulfill service requirements to guarantee reliable services.

It is possible that graph *G*, representing the network, becomes disconnected after experiencing a failure. In this case, G fragments into multiple components or subgraphs, preventing nodes within these components from communicating with one another. Therefore, if sender and receiver nodes are in separate components, packet delivery between them will be impossible, regardless of the employed network design.

Two scenarios can result in unsuccessful packet flow delivery upon a failure event: (a) unavailability due to delayed re-routing after failure (i.e., delay-convergence problem) or (b) unavailability caused by unreachability when sender and receiver nodes reside in different components (i.e., host pair). Availability and reachability are closely linked in network analysis: while availability evaluates the network's current state, unreachability investigates potential future dysfunction.

Figure 2 demonstrates packet routing in a network. This example features two host nodes (*P*2, *P*4) and a number of other devices, such as switches connecting them, with each link bearing a cost used to establish the shortest path for packet routing. As shown, host nodes *P*2 and *P*4 remain reachable after the *R*2 and *A*2 link failure, as an alternative path exists through *R*1. Nevertheless, the unavailability resulting from delays in the recovery process—which involves detecting a failure event and computing an alternative—is associated with the internal operational capabilities of the network.

*Characterizing Network Unreachability*

In this subsection, we use network unreachability analysis to identify the bottlenecks that affect the operational lifetime of packet transmission and reception in a network. We demonstrate how graph dynamicity changes over time and how failure events impact the availability of different network components. To illustrate network unreachability, we introduce the following auxiliary graph notations:

- The degree of node *v* in graph *G*, denoted by $deg(v)$, represents the number of links connected to *v*.
- The cut-edge set $e\_cut(G)$ for a connected graph *G* is a collection of links whose removal results in the disconnection of *G*. This set is not unique. The minimum cut-edge set, $min\_e\_cut(G)$, comprises the cut-edge set with the fewest links. For any connected graph *G*, it holds that

$$|min\_e\_cut(G)| \leq min\_deg(G) \tag{1}$$

where $|.|$ signifies the size of a set, and $min\_deg(G)$ represents the minimum node degree in *G*. Network disconnection leads to packet flow disruption, which can be evaluated by measuring the number of fragmented components and their sizes (i.e., the number of nodes in each component). A higher number of components indicates increased degradation in packet flow delivery and thus greater unreachability among network elements. Suppose a set of links $\aleph \subseteq L$ is removed from graph *G*, resulting in network *G*-$\aleph$. Let *C(G-$\aleph$)* denote the number of components in the *(G-$\aleph$)* graph. The main component of *(G-$\aleph$)*, referred to as M(*G*-$\aleph$), possesses the highest number of nodes. If C(*G*-$\aleph$) = *i + 1*, the minimum set of $\aleph$ is called the $i^{th}$-order cut-edge set of *G*. As such, the principal minimum cut-edge sets contribute significantly to network unreachability. Hence, path diversity is an essential factor that improves network availability by providing multiple routes for traffic delivery when rerouting is required due to failures.

## 5. Problem Formulation

Consider a graph $G$ that represents a communication network, with $k$-terminal pairs denoted $(s_k, t)$, $1 \leq k \leq n$ through which traffic flows are anticipated to move from $s_k$ to $t$. The set of all potential paths connecting these terminal pairs, $\aleph$, is represented by $\{p_1, p_2, \ldots, p_r\}$. Each path, $p_i$, in $\aleph$ connects the $(s_k, t)$ pair and is arranged in ascending order based on its length, such as $p_1 \leq p_2 \leq \ldots \leq p_r$, where $1 \leq r \leq \sum_{j=0}^{n-2} (n-2j)j!$. The upper bound of $r$ represents the size of set $\aleph$ when $G$ is a complete graph comprising $n$ vertices. The goal is to find a single set of $p_i$ so that the maximum link utilization is minimized. Subsequently, this set should be divided into two edge-disjoint multipaths: working and backup. The ideal outcome involves both multipaths having larger and comparable sizes for enhanced robustness and performance.

## 6. The Proposed Availability Framework

Mixed Integer Linear Programming (MILP) solvers, such as CPLEX [12], demonstrate the capacity to effectively tackle complex combinatorial problems with limited sizes. Nonetheless, when it comes to addressing large-scale problems, these solvers frequently encounter computational intractability, thus failing to generate optimal solutions. To counteract this inherent hardness, our proposed framework strategically combines two distinct methodologies, namely optimization and heuristic techniques, to bridge the gap between computational limitations and problem-solving efficacy.

### 6.1. Phase One: MILP Formulation for EDP

In this phase, we aim to develop a Mixed Integer Linear Programming (MILP) model that determines an optimal single multipath to route traffic from multiple source nodes ($k$) to a specific destination node. The aim is to minimize the maximum link utilization, enhancing overall network efficiency.

- Input to the optimization problem

The arc data structure is composed of four key components: the originating node (fromNode), destination node (toNode), capacity, and cost. The available bandwidth or capacity of a given arc between nodes $(i, j)$ is referred to by $c_{ij} \in \mathbb{R}^+ \cup \{0\}$. In addition, we denote the cost associated with an arc $(i, j)$ as $d_{ij} \in \mathbb{R}^+ \cup \{0\}$. Moreover, we define $D$ as a demand matrix, in which the element $D(s, t)$ specifies the traffic flow demand between a pair $(s, t) \in N \times N$.

- Decision Variables

The variable $f^t_{(i,j)}$ denotes the amount of traffic that traverses arc $(i, j)$ directed towards node $t$. We use the term $u_{ij}$ to represent the accumulated traffic loads' overall demands that pass through the arc $(i, j)$. The Boolean decision variable $\beta^t_{(i,j)}$ specifies whether or not any traffic will be traversed through the arc $(i, j)$ directed towards node $t$. Additionally, the distance concerning TE metric cost from a node $j$ to another node $t$ is denoted by $l^t_j$. The value $z \in \mathbb{R}^+ \cup \{0\}$ signifies the maximum utilization of links across the entire network.

- Validity and Continuity

To maintain the desired traffic flow and guarantee that the required demand is transferred from node $s$ to node $t$, the following constraints have been established for $\forall\, i, t \in N$. These constraints optimize traffic management and ensure efficient traffic flow between the designated nodes.

$$\sum_{\substack{(i,j)\in LD \\ i=t}} f^t_{(i,j)} - \sum_{\substack{(j,k)\in LD \\ k=t}} f^t_{(j,k)} = -\sum_{k\in N} D(k,t) \tag{2}$$

$$\sum_{(i,j)\in LD} f^t_{(i,j)} - \sum_{(j,i)\in LD} f^t_{(j,i)} = D(i,t) \,,\ i \neq t \tag{3}$$

A further consideration for maintaining validity and continuity within the network is to ensure that the amount of data flow transmitted across any given arc does not exceed its maximum capacity, represented by its available bandwidth. This particular constraint can be expressed mathematically as follows.

$$\sum_{t \in N} f^t_{(i,j)} \leq c_{ij}, \ \forall (i,j) \in LD \tag{4}$$

- Bandwidth and Length Constraints

$$f^t_{(i,j)} \leq \sum_{k \in N} D(k,t) \ if \beta^t_{(i,j)} = 1, \ t \in N, \ (i,j) \in LD \tag{5}$$

The given constraint indicates that the total traffic load directed to node t and passing through the arc $(i, j)$ must not exceed the cumulative demands directed towards t. This applies when the arc $(i, j)$ has been selected for traffic delivery. To maintain candidate paths of the multipath as short as possible, we have established the following constraints.

$$0 \leq l^t_j + d_{ij} - l^t_i \tag{6}$$

$$l^t_j + d_{ij} - l^t_i \leq \left(1 - \beta^t_{(i,j)}\right) \tag{7}$$

$$\left(1 - \beta^t_{(i,j)}\right) \leq l^t_j + d_{ij} - l^t_i \tag{8}$$

- Utilization

$$f^t_{(i,j)} \leq z * c_{ij}, \ \forall (i,j) \in LD, \ t \in N \tag{9}$$

This constraint maximizes link utilization.

- Complete Optimization Formulation

The comprehensive MILP formulation can be summarized as follows:

$$\min z$$
$$\text{subject to}: \ (2) - (9)$$

### 6.2. Phase Two: Finding Working and Backup Multipaths

Dividing the outcome of our MILP into two separate multipaths is a complex task, as our primary goal is to maximize the sizes of both multipaths. In this subsection, we introduce an innovative algorithm, LDCR, which effectively partitions the MILP-generated multipath into a pair of edge-disjoint multipaths. The fundamental concept behind the LDCR algorithm incorporates two key elements: distributing the links connected to source nodes between active and backup multipaths and addressing any conflicts between active and backup paths that arise over mutual links. Algorithm 1 shows these key elements.

In the initial step, the first-mile links are allocated between the two multiple paths (i.e., working (W) and backup (B)) to ensure fairness distribution. To accomplish this, we determine the frequency of each link's occurrence in the outcome of our MILP, $G'$, sort them in descending order, and then systematically allocate them across the W and B multipaths/sets, as illustrated in lines 1–4.

In Line 5, we assign the $G'$ paths between W and B sets according to their connectivity to the links adjacent to source nodes which we already distributed in the previous step.

In the second step, LDCR effectively resolves conflicts related to shared links between W and B sets. The find_intersection function, as seen in Lines 7–10, determines the common links between working and backup multipaths.

This function comprises multiple loops designed to traverse each path within both the working and backup multipaths, subsequently examining for overlaps. Instances where two paths—one from W and another from B—share a common link are identified

as conflicting paths. To resolve such conflicts, one of the paths must be eliminated. The path to be removed is determined by evaluating the extent of its conflicts with other paths originating from the opposite set; the greater the conflict prevalence, the higher the likelihood of removal. This process is comprehensively delineated in lines 12–24.

---

**Algorithm 1: Link Distribution and Conflict Resolution (LDCR)**

---

**Input:** single multipath $G'$ from phase one, src, and des.
**Output:** two disjoint multipaths: working and backup

1　　**Initialize sets for working (W) and backup (B) multipaths**.
2　　Calculate the first-mile links using MILP, resulting in G'.
3　　Sort the links in G' by frequency in descending order.
4　　For each link in the sorted list, allocate the link to either W or B:

　　o　　If W size < B size: Add the link to W set.
　　o　　Else: Add the link to B set.

5　　Assign the paths from G' to W and B sets:

　　o　　Iterate through each path from G'.
　　o　　If a path is connected to a source node with an allocated link in W: Add the path to W set.
　　o　　Else: Add the path to B set.

6　　**Using find_intersection function, find shared links between W and B sets:**
7　　Initialize a list for shared_links.
8　　For each path_w in W:
9　　For each path_b in B:
10　Traverse both paths, and if they share a common link, add it to shared_links.
11　**Resolve conflicts caused by shared links:**
12　For each conflict_link in shared_links:
13　Initialize conflict_count_w for working path and conflict_count_b for backup path.
14　For each other_link in shared_links (excluding conflict_link):
15　Increment conflict_count_w if other_link appears in the same path as conflict_link in W set.
16　Increment conflict_count_b if other_link appears in the same path as conflict_link in B set.
17　Determine which path to remove based on higher conflict_count:
18　If conflict_count_w > conflict_count_b:
19　Remove the path containing conflict_link from W set.
20　Else if conflict_count_b > conflict_count_w:
21　Remove the path containing conflict_link from B set.
22　Else:
23　Choose a random path among the conflicting paths with equal conflict_counts and remove it.
24　Repeat steps 6–11 until no shared links are found between W and B sets.

---

While the LDCR algorithm effectively identifies link-disjoint working and backup multipaths through a two-stage strategy, a potential exists to enhance the number of paths within these two sets. In the following subsection, we introduce an advanced version of the algorithm, called Enhanced LDCR (E-LDCR), which substantially augments the overall performance in terms of incorporating a greater number of paths in both working and backup multipaths, hence elevating the network availability.

- E-LDCR Methodology

Maximizing the number of paths in both working and backup systems necessitates a thorough assessment of the potential loss and gain associated with removing conflicting paths. To achieve this, we introduce an advanced method that utilizes three variables: release (r), sacrifice (s), and keep (k). Each variable represents the number of removed conflicting paths, the number of links no longer shared, and the retained conflicting paths that no longer pose conflicts, respectively. By incorporating these values into a single formula, we can generate a Key Indicator (KI) that identifies which paths should be

prioritized for removal. This KI, referred to as RSK, offers an effective approach to managing path conflicts effectively.

$$RSK = \frac{r+k}{s} \tag{10}$$

A higher *RSK* value signifies an increased likelihood of eliminating conflicting paths associated with it. We will now illustrate the way *RSK* operates. Figure 3 depicts a single multipath consisting of 28 individual paths as a result of the MILP solution.

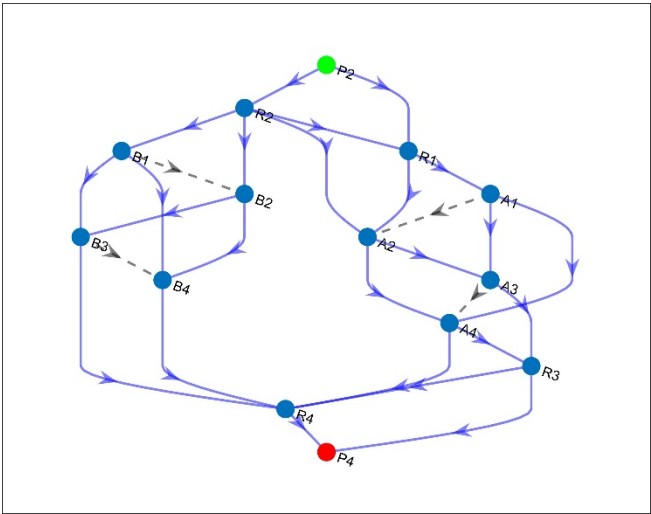

**Figure 3.** An illustration of the MILP single multipath. The dotted lines represent connections not utilized in the initial $G'$ multipath.

Figure 4 shows the LDCR/E-LDCR assignment of links connected to source P2 as either working (represented in blue) or backup (represented in red) multipaths. Note that dashed lines are not a part of these multipaths.

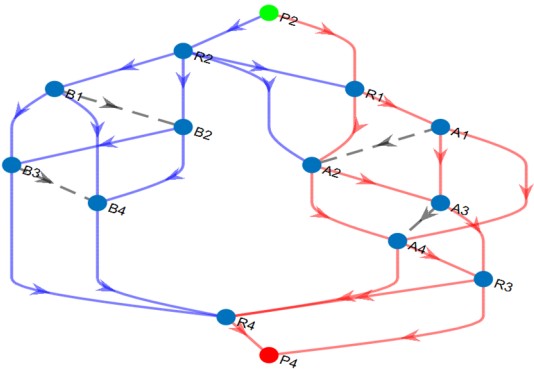

**Figure 4.** An illustration of the MILP single multipath after the first stage of LDCR /E-LDCR. (R2, R1) and (R2, A2) links are deemed incompatible due to E-LDCR's conflict resolution.

Table 1 shows all conflict links and the conflicting paths within the working and backup multipaths presented for the system in Figure 4. Note that each multipath features unique indexing for its paths. For instance, path index 1 in the second column should not be confused with path index 1 in the third column; the former pertains to the working multipath, while the latter refers to its backup counterpart. This distinction is crucial for accurately understanding the data in Table 1 concerning the relationships between conflicting paths in both working and backup multipaths.

**Table 1.** *RSK* required parameters—the number of shared links, the conflicting working paths, and the conflicting backup paths for each shared link.

| Links of Conflict (Index, Name) | Conflicting Working Paths (Path Index) | Conflicting Backup Paths (Path Index) |
|---|---|---|
| 7, (R1, A2) | 1 | 1, 2 |
| 8, (R1, A1) | 2 | 3, 4 |
| 11, (A2, A4) | 1, 3 | 2 |
| 13, (A1, A4) | 2 | 4 |

To enhance the visual representation of the proposed method, Table 2 contains the actual sequence for every conflicting path.

**Table 2.** The indexes and actual conflicting paths for both working and backup multipaths.

| | Path Index | Path |
|---|---|---|
| **Working** | 1 | P2—R2—R1—A2—A4—R4—P4 |
| | 2 | P2—R2—R1—A1—A4—R4—P4 |
| | 3 | P2—R2—A2–A4—R4—P4 |
| **Backup** | 1 | P2—R1—A2—A3—R3—P4 |
| | 2 | P2—R1—A2—A4—R3—P4 |
| | 3 | P2—R1—A1—A3—R3—P4 |
| | 4 | P2—R1—A1—A4—R3—P4 |

We define the contribution of a specific conflicting path within a multipath (working or backup) towards a shared link $(i, j)$ as the number of paths included in the opposing multipath (backup or working) that share the link $(i, j)$. To illustrate this concept, let us examine Table 1, where a link with index 8 (represented as R1—A1) is part of three conflicting paths: one with index 2 in the working multipath and two others with indices 3 and 4 in the backup multipath. Consequently, the contribution of the path with index 2 in the working multipath towards the conflict concerning link R1-A1 is due to the presence of two corresponding paths in the backup multipath. However, for paths 3 and 4, their contribution to this conflict is limited to one.

Next, we compute $r$, $s$, $k$ variables to ascertain the *RSK* value for each potential path removal. Table 3 demonstrates the impact of eliminating conflicting paths in a sequential order to address the conflict. In this Table, *RSK* variables are calculated using data from Table 1. The removal order strategy targets each shared link and evaluates $r$, $s$, $k$ variables for two scenarios: (1) the removal of working conflicting paths and (2) the removal of backup paths. In both instances, we employ the W- or B- naming convention to signify working and backup multipaths, followed by the paths' indices. For instance, Table 3 presents two cases for resolving the conflict over link (A2, A4): W-1,3, which implies removing working paths with indices 1 and 3, and B-2, signifying the removal of backup paths with index 1. We will use this example (for link (A2, A4)) to illustrate how to calculate $r$, $s$, $k$ in both cases.

Case 1 (W-1,3): By eliminating conflicting working paths 1 and 3, two paths (specifically, paths 1 and 3) will be unavoidably sacrificed, resulting in a value of s equal to two. This action releases two shared links, as not only link (A2, A4) is freed, but also link (R1, A2), since working path 1 is the sole conflicting path over link (R1, A1). Consequently, r equals two. Furthermore, $k$ equals two as backup paths 1 and 2 are preserved.

The maximum *RSK* value is four, achieved by eliminating W-2. Subsequently, the process is recalculated to remove the next highest *RSK* value until all conflicts are addressed. Figure 5 depicts the outcome of the E-LDCR algorithm.

**Table 3.** Calculating the *RSK* variables out of Table 1.

| Variable/Path Removal | | Release ($r$) | Sacrifice ($s$) | Keep ($k$) | *RSK* |
|---|---|---|---|---|---|
| (R1, A2) | W-1 | 1 | 1 | 2 | 3 |
| | B-1,2 | 2 | 2 | 2 | 2 |
| (R1, A1) | W-2 | 2 | 1 | 2 | 4 |
| | B-3,4 | 2 | 2 | 1 | 1.5 |
| (A2, A4) | W-1,3 | 2 | 2 | 2 | 2 |
| | B-2 | 1 | 1 | 2 | 3 |
| (A1, A4) | W-2 | 2 | 1 | 2 | 4 |
| | B-4 | 1 | 1 | 1 | 2 |

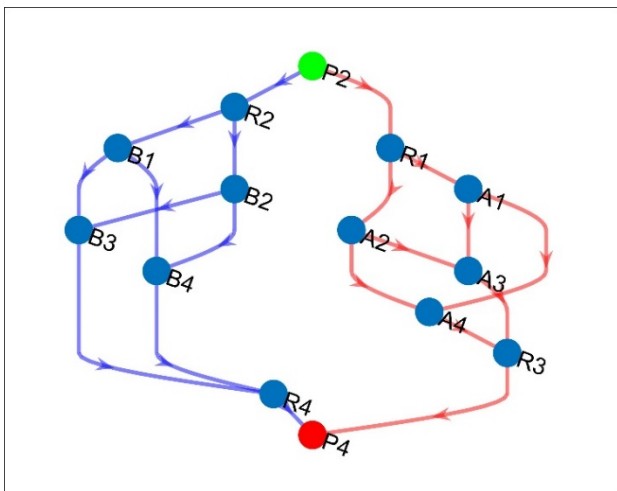

**Figure 5.** Two edge-disjoint multipaths: blue for working (W) and red for backup (B).

Figure 5 illustrates the elimination of paths containing shared links (R2, R1), (R2, A2), (A4, R4), and (R3, R4). Consequently, four remaining paths are present for each multipath. In contrast, the original LDCR generated five and two paths for working and backup multipaths. E-LDCR offers a more balanced distribution and enhances the number of these paths, affecting the network's overall availability.

*6.3. Qualitative Analysis of the Working and Backup Sets*

In this subsection, we conduct a comparative analysis of the working and backup multipaths. To facilitate this comparison, we employ several metrics, including edge-cut sets, an "influence" measure encompassing connectivity and betweenness centrality, and the operational probability of paths within each multipath.

6.3.1. The Edge-Cut Set: Revisited

Network disconnection can negatively impact the delivery of packet flows. This disruption can be evaluated by examining the number of fragmented components and their corresponding sizes (specifically, the number of nodes within each component). An increase in fragmentations results in further degradation of packet flow delivery.

As illustrated in Section IV (A), $e\_cut(G)$ is a collection of edges whose removal results in the disconnection of $G$. Furthermore, the minimal edge-cut set, $S = min\_e\_cut(G)$, is a specific set of edges that adhere to the following criteria: (i) the removal of all edges present in $S$ causes $G$ to be divided into two distinct connected components, namely $X_S$ and $\underline{X_S}$, and (ii) no proper subset of $S$ fulfills condition (i) with a fewer number of edges. As a result, the minimal edge-cut set leaves node $s$ in $X_S$ and node $t$ in $\underline{X_S}$ for pair terminal

$s, t \in G$. Utilizing the principle of edge-cut sets, we shall evaluate the efficacy of network availability metrics' working and backup multipaths.

### 6.3.2. The Influence Measure

We present the "influence" metric, $I$, as a fusion of graph $G's$ connectivity and the betweenness centrality of each individual node. The $n \times n$ influence matrix is defined, in which the element $I(i, j)$ represents the impact of node $i$ on node $j$.

$$I = AB \tag{11}$$

where $A$ is the $n \times n$ adjacency matrix of the graph $G$, and the $(i, j)$ element of $A$ is given by

$$a_{ij} = \begin{cases} 1, & if \ there \ is \ a \ link \ between \ node \ i \ and \ j \\ 0, & otherwise \end{cases} \tag{12}$$

And $B$ is an $n \times n$ diagonal matrix; the $(i, i)$ element of $B$ is given by

$$b_{ii} = \sum_{s,t \neq i} \frac{p_{st}(i)}{p_{st}} \tag{13}$$

where $p_{st}$ represent the total number of shortest paths connecting node $s$ to node $t$, and $p_{st}(i)$ denote the number of these paths that traverse through arc $(i, j)$, where $j$ is an adjacent node to $i$. A higher value of $b_{ii}$ implies that a greater number of paths pass through arc $(i, j)$, which subsequently indicates that node $i$ exerts significant control over the network availability. Moreover, a larger $I(i, j)$ value indicates that node $i$ possesses a stronger influence on node $j$. Consequently, arc $(i, j)$ has a higher criticality within its multipath network.

Next, we analyze the failure probability distribution of links. This helps us to obtain a deeper understanding of those links of significant importance and exhibit a high probability of failure. Consequently, this allows identifying the accurate critical links present in both working and backup multipaths. Driven by the evaluation of these multipaths' criticality—achieved using availability and betweenness metrics—we propose a novel K-means-based clustering algorithm designed to partition our MILP multipath.

### 6.3.3. Path Operational Probability

Let $p_j$ represent a path connecting two terminal nodes, $s$ and $t$. A path $p_j$ is considered operational if all its constituent links are present. The probability of $p_j$ being operational, denoted as $Pr(p_j)$, can be expressed by the subsequent formula:

$$Pr\left(p_j\right) = \prod_{1 \leq k \leq |p_j|, \ e_k \in p_j} Pr(e_k) \tag{14}$$

Let $p_{j\_usd}$ represent the event that routing path $p_j$ is being used for transferring packets from source $(s)$ to destination $(t)$. The probability of this event is expressed as follows:

$$Pr\left(p_{j\_usd}\right) = Pr\left(p_j\right) \prod_{\alpha=2}^{j-1} \left(1 - Pr\left(p_\alpha\right)\right) \tag{15}$$

where $\alpha$ holds the length of alternative paths, in the backup set, connecting the two terminal nodes. As indicated in Section 6.3.1, a link $(i, j)$ is an edge cut if no alternative paths are available for re-establishing a connection between nodes $i$ and $j$.

Equation (14) offers an advanced metric for assessing the performance of paths within both working and backup multipaths. Moreover, when all paths in a working multipath are deemed inadequate for traffic routing, Equation (15) presents valuable guidance on when a network operator should utilize the backup multipath.

## 7. Experimental Results

In this section, we demonstrate the validation and effectiveness of our proposed methodology in terms of networkwide cost and availability performance metrics. To validate our approach, we perform experiments using the network depicted in Figure 6, consisting of 34 nodes and 82 edges. This topology signifies a User Plane Function (UPF) architecture, a crucial 3GPP 5G core infrastructure component. Figure 6 shows ten sites (0–9), including eight anycast groups (0–7). Sites 8 and 9 serve as source and destination nodes. Each anycast group comprises a complete graph with four nodes, i.e., K4. All links connecting any two sites feature the same TE metric, set at 50. In contrast, the TE metric for every link within a site is set to 10. We utilize MATLAB R2022b for all simulations conducted on an x64-based Intel Core i7-1165G7 CPU @ 2.80GHz with 16 GB of RAM, running Windows 11.

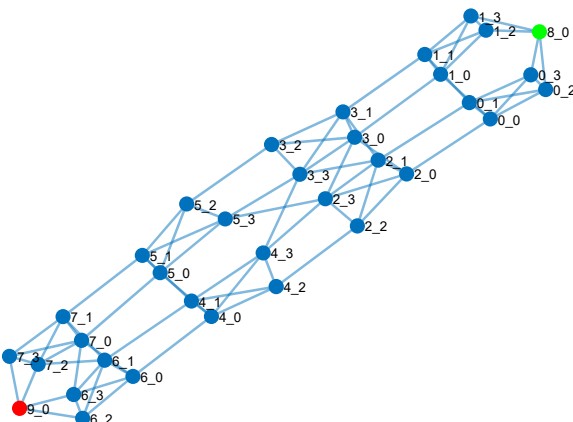

**Figure 6.** Network topology with 10 sites including 8 anycast groups. The notation a_b signifies node b belonging to anycast group a. The data flow originates from the green node and concludes at the red node.

### 7.1. Validation

In this experiment, the data traffic of 20 units was transferred from node 8_0 to node 9_0. The solution for our MILP corresponding to the network illustrated in Figure 6 is represented by the entire graph depicted in Figure 7. An even distribution of traffic can be observed among the ECMP routes. As expected, the MILP effectively balanced the network's traffic load, resulting in a symmetrical output in the multipath routing. To further divide this multipath, our E-LDCR algorithm is employed. The final outcome is also presented in Figure 7, where the working multipath appears in blue, while its backup is in red.

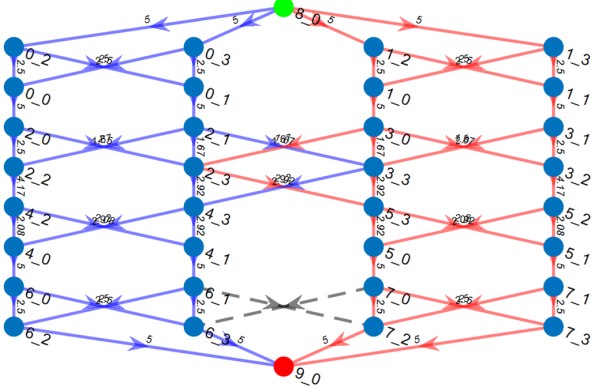

**Figure 7.** The MILP-based multipath is split into two edge-disjoint multipaths: working in blue and backup in red. The label on arcs refers to the load passing over this arc. Dashed links carry no traffic.

The number of individual paths in the original MILP multipath is 160; the first links' assignment divided the paths evenly among the working and backup multipaths. The result shows 40 paths for each.

### 7.2. Effectiveness

Our proposed framework is compared to the widely recognized inverse-capacity OSPF (InvCap-OSPF) routing solution, which allocates link weights inversely proportional to their capacities, facilitating efficient routing choices. As demonstrated in Figure 8, we present a network topology evaluation, highlighting our MILP result (exhibited in red) pertaining to one-unit demands between four source nodes (displayed in green) and a single destination node (indicated as G in red). The links are labeled as traffic-load units/capacity/weight triplets.

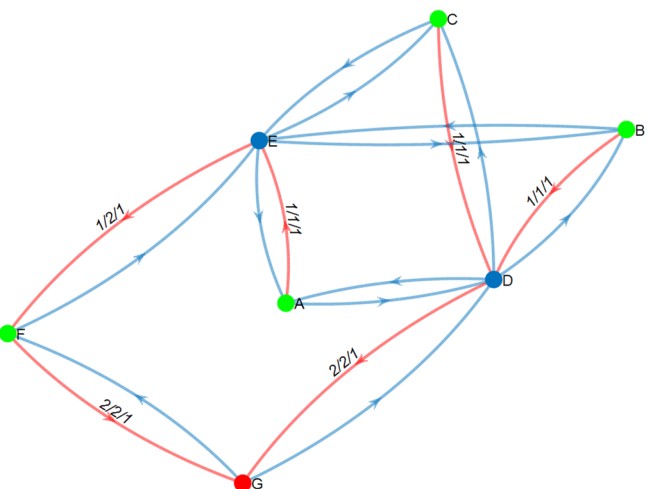

**Figure 8.** Our MILP solution, illustrated in red, represents a one-unit demand between four source nodes, depicted in green, and a single destination node, G highlighted in red. Each link is labeled with a triplet consisting of traffic-load units, capacity, and weight.

On the other hand, InvCap-OSPF assigns uniform weight values of one to all links except the (D, G), (E, F), and (F, G) links. These specific links receive half the weight due to their doubled capacity. The systematic methodology introduced in our framework enables a smooth integration of the routing structure, effectively accommodating diverse network topologies and loads, leading to higher network availability.

### 7.2.1. Networkwide Cost Function

In the experiment, we adopt the link cost function proposed in [24] to evaluate the performance of our MILP solution in relation to InvCap-OSPF. As a function of utilization, link cost increases with exponential growth when utilization surpasses 100%. The collective cost of a routing strategy is determined by aggregating the individual costs of all links. To execute our experiment, we apply the network and weight configurations depicted in Figure 8, which illustrates a traffic load demand valued at one between the source nodes (A, B, C, and F) and destination node G. The networkwide cost computation entails using the following 12 traffic load values [0, 0.2, 0.4, 0.6, 0.8, 1.0, 1.2, 1.4, 1.6, 1.8, 2.0, 2.2]. Figure 9 presents a comparison between our proposed framework and InvCap-OSPF outcomes with each data point reflecting the networkwide cost accrued at its respective load value.

Figure 9 illustrates that InvCap-OSPF leads to network overload in the initial phase of iterations across various traffic load values. Conversely, our proposed MILP demonstrates a significantly improved networkwide cost performance. It can accommodate approximately 80% more traffic load than InvCap-OSPF before reaching a cost value of one. This outperformance is due to our proposed solution's ability to efficiently distribute traffic

flows, ensuring network equilibrium and minimizing the maximum utilization of links. Consequently, penalties for utilizations exceeding 100% are effectively avoided.

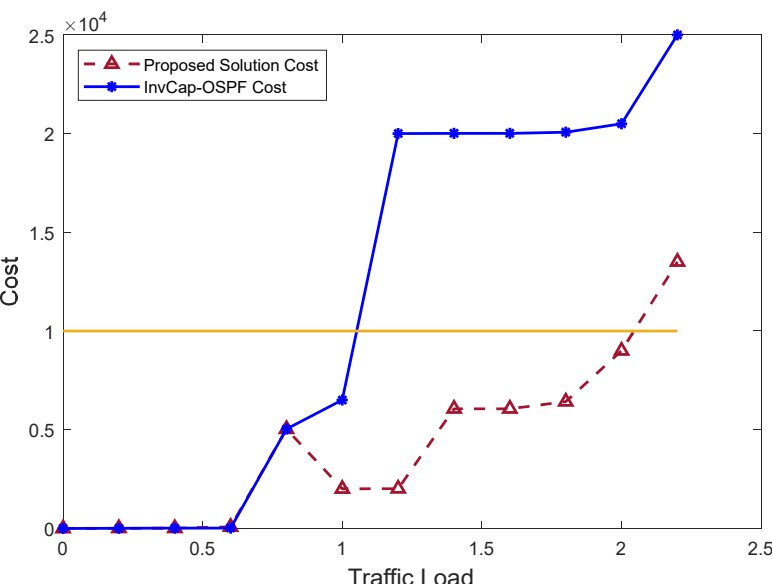

**Figure 9.** Networkwide cost for our MILP solution versus InvCap-OSPF.

### 7.2.2. Network Availability

This experiment demonstrates the outperformance of E-LDCR in comparison to LDCR concerning network availability. We assess the interruption of traffic load/service during controlled faults when traffic loads are conveyed between (*s,t*)-pair terminals utilizing the multipath produced by both LDCR and E-LDCR.

A successful service delivery between any (*s,t*)-pair nodes via either working or backup multipath is assigned a value of one, indicating high availability; otherwise, it is deemed zero. For this analysis, a network analogous to the one depicted in Figure 6 is employed. The network contains 30 anycast groups connected by a connectivity density parameter, densPara = 0.5. Random C pairs from these groups are interconnected, where C is the maximum number such that C < NumPairs × densPara.

In this case, C = 2017, $n$ = 120, and $m$ = 2104. Two arbitrary nodes from each C anycast pair are chosen and linked so that each node connects to both nodes in the other anycast pair (i.e., following a complete bipartite pattern). A failure probability value is randomly assigned to each edge, reiterating this assignment process 30 times. During each iteration, the 20 edges possessing the highest failure probability will be eliminated one by one. The availability will be computed for every edge removal and the average derived from these 30 values will be determined. In this experiment, two (*s,t*)-pair nodes are chosen, with one designated for each multipath. The marked points in Figure 10 denote these average values.

As illustrated in Figure 10, E-LDCR performs better in ensuring network availability. In the initial phase of the failure process, the availability values exhibit marginal superiority for E-LDCR over standard LDCR. This is due to the network's retention of most individual paths at this stage. As link failures accumulate, E-LDCR alleviates their effects and maintains network availability at approximately 47% higher than the performance observed under LDCR. In the final stages of the failure process, the availability under the LDCR-based network declines more rapidly compared to that under E-LDCR. This can be attributed to the presence of additional individual paths within E-LDCR capable of managing increased failures while continuing to accommodate traffic loads.

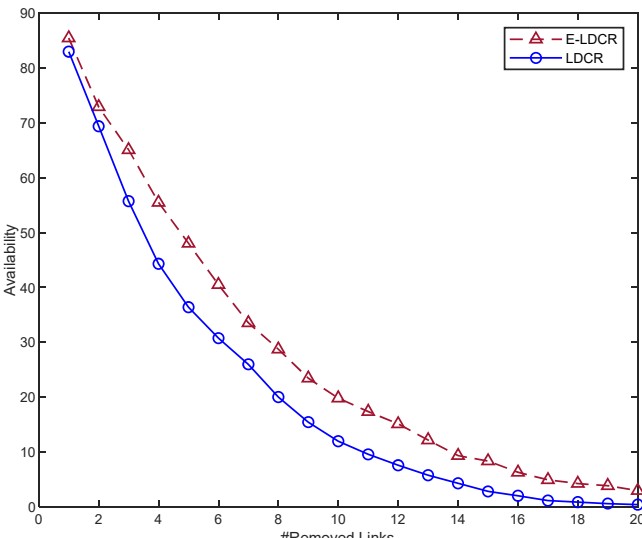

**Figure 10.** Network availability under successive link failure: LDCR vs. E-LDCR.

### 7.2.3. Max-Utilization Performance Evaluation

In the following experiment, we utilize the network model illustrated in Figure 11, where each of the nodes A, B, C, and F transmit one traffic unit towards destination G. The capacity for all links connected to nodes A, B, and C is set to one, while the remaining links (D, G), (E, F), and (F, G) feature a capacity of two. Utilizing Unit-OSPF, which assigns a weight of one to all links, results in maximum utilization at link (D, G) with a load of three units and a capacity of two, thus reaching 150% utilization. Conversely, InvCap-OSPF adjusts the weights of links (D, G), (E, F), and (F, G) to $\frac{1}{2}$, as discussed earlier in the effectiveness subsection. This modification does not impact traffic routing from nodes A, B, and C since node D exists in the label/SID stacks of routing path commencing these nodes to the destination node G. Nonetheless, as maximum utilization continues to stand at 150%, InvCap-OSPF offers no improvement over Unit-OSPF.

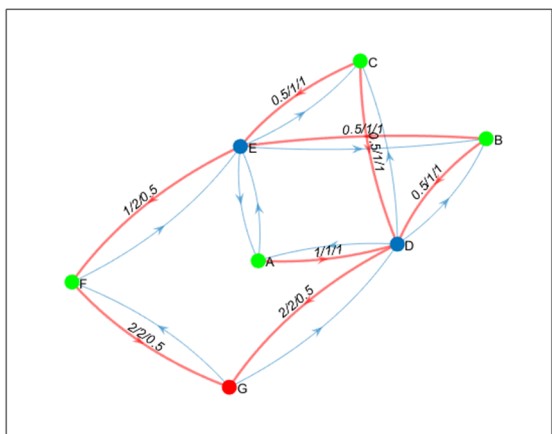

**Figure 11.** Our MILP result when setting the traffic-load units of links (C, E), (B, E), (C, D), and (C, E) to $\frac{1}{2}$. Links labeled as triplet traffic-load units/capacity/weight.

On the other hand, our proposed solution in Figure 11 attains optimal utilization at 100%. This utilization level occurs on links (D, G) and (F, G), each carrying corresponding loads and capacities of two units. Link (A, D) also exhibits ideal utilization with one load unit and an identical capacity. It is worth noting that links (B, D), (B, E), (C, D), and (C, E) experience 50% utilization aimed at optimizing load balancing while minimizing peak utilization. As a result of this approach, our solution can accommodate a demand increase

of up to 50%, presenting a substantially more desirable alternative for network operators than deploying supplementary links.

### 7.3. Evaluating the Quality of Working and Backup Multipaths
### 7.3.1. Operational Probability

As depicted in Figure 12, the working multipath consists of three paths, represented in blue, while the backup multipath has two paths, indicated in red. Following an exponential distribution model, each connecting edge possesses a distinct failure probability determined through a random selection process.

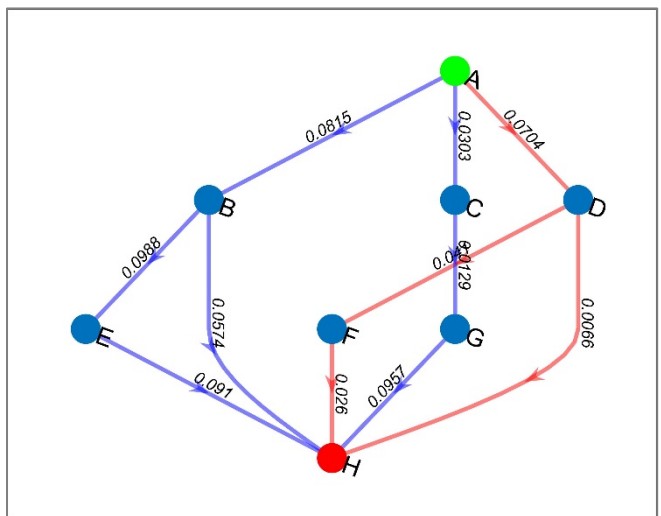

**Figure 12.** Two edge-disjoint multipaths: working and backup in blue/red color. Each link has associated probability of failure.

Table 4 demonstrates the utilization of Equations (14) and (15). As an example, the operational probability of the ABEH working path can be determined using Equation (14) as follows.

$$\Pr(ABEH) = \Pr(AB) \times \Pr(BE) \times \Pr(EH)$$
$$= (1 - 0.0815) \times (1 - 0.0988) \times (1 - 0.091) = 0.75$$

**Table 4.** An illustration of the operational probability of both primary and backup multipaths and the probability that the backup route will be employed in case the primary one fails.

| Multipaths | | Operational Probability | Normalized Operational Probability | Probability of Using Backup Multipath if the Working Is Failed |
|---|---|---|---|---|
| Working | ABEH | 0.75 | 0.18 | |
| | ABH | 0.87 | 0.20 | |
| | ACGH | 0.87 | 0.20 | |
| Backup | ADFH | 0.86 | 0.20 | 0.0039 + 0.0041 = 0.008 |
| | ADH | 0.92 | 0.22 | |

Both the working and backup multipaths exhibit high operational probabilities for their respective paths, with a minimum of 0.75 for ABEH and a maximum of 0.92 for ADH. Nevertheless, it is improbable that the network operator will utilize the backup multipath anytime soon, since three paths within the working multipath already have high operational probabilities, making the use of the backup multipath less likely.

### 7.3.2. Graph-Related Metrics

As illustrated in the subsection on operational probability, we employ Figure 12 as a foundation for the comparative evaluation in accordance with graph-related metrics such as nodal degree, edge-cut set, and influence indicator. The findings are presented in Table 5.

**Table 5.** Statistical graph-theoretic measures for working and backup multipaths.

| Metric | Working | Backup |
|:---:|:---:|:---:|
| $AvgDeg(G)$ | 2 | 1.75 |
| $min\_e\_cut(G)$ | $(A, C)$ <br> $(C, G)$ | $(A, D)$ |
| $I$ | $\frac{1}{3}$ for C, G, and E <br> $\frac{2}{3}$ for B | 1 for D <br> $\frac{1}{2}$ for F |

The metrics of nodal degree, influence indicator, and cut links offer valuable insights into the graph's overall structure. Table 5 shows that the working multipath demonstrates superior performance concerning average nodal degree and minimum edge cut. In terms of the influence indicator, the backup displays a higher risk for failure as the intermediate node D possesses a greater $I$ value compared to its counterpart within the working multipath. If D or any of its connected links fail, the backup multipath availability will be highly impacted as the network will be unable to facilitate packet delivery from node A to node H.

### 8. Conclusions

This paper presents a comprehensive framework for effective network traffic flow management irrespective of the routing paradigm. The framework consists of two primary phases: first, determining the optimal routing of multi-source single-destination traffic flows through the formulation of a MILP problem to minimize maximum link utilization; and second, establishing edge-disjoint multipaths using our designed algorithms, LDCR and E-LDCR. Moreover, we investigate the qualitative aspects of working and backup multipaths regarding edge-cut sets, centrality measures, and operational paths. Our simulation results demonstrate the significant improvements achieved by our proposed solution compared to the InvCap-OSPF routing solution, with an 80% increase in traffic load handling capacity and a 47% higher network availability.

Future work could involve further enhancements to the E-LDCR algorithm, investigating the framework's scalability for larger and more complex network environments, and expanding the focus to explore possible applications in other domains, such as IoT and smart infrastructures.

**Funding:** This research is supported by a grant from the University of Winnipeg under grant number 31996.

**Data Availability Statement:** Not applicable.

**Conflicts of Interest:** The authors declare no conflict of interest.

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
