# Peer review of "Enhancing Network Availability: An Optimization Approach"

_computation, doi:10.3390/computation11100202_

Round 1

Reviewer 1 Report

The paper proposes a three-phase disjoint-path framework that improves availability by directing traffic flows through separate paths. The framework includes the Optimization phase for identifying optimal multipath solutions, the Path Separation phase for dividing the multipath into working and backup sets and the Quality Assessment phase for evaluating the robustness of both sets.  Simulations demonstrate the proposed framework's validation and effectiveness in enhancing network availability. The results show significant improvements compared to the InvCap-OSPF routing solution. The authors have planned further work to investigate the framework's scalability for larger and more complex network environments and to explore applications for IoT and smart infrastructures.

The manuscript is well written. It presents the proposed framework in detail and the results of various phases of the method are demonstrated with examples using small network topologies.

I have the following comments (and also some lines to be checked):

1) I think that the first paragraph of the Introduction section is too technical (compared to the second paragraph, for example).

2) Some abbreviations are used before they are defined (line 100: ILP, line 103: MILP,...). (In addition, it is also possible to add a list of abbreviations at the end of the paper.)

3) segment Identifier (SID) -> Segment Identifier (SID)? (Please check this)

4) Line 104: K segment (Please check these and check also the use of italics of the variables in the paper.)

5) Lines 290, 291: Please check equations (7) and (8).

6) Figure 6: You could explain the notation here. Figure 7 has too small and overlapping numbers (Please check the figures of the paper).

7) Indentation of some equations. Two Equation 3 lines 430 and 448.

8) Some tables have too small fonts (Tables III, IV).

9) Lines 6,..., 625: "...simulations demonstrate...". Later, in the text and in the Conclusions section "Our experimental results...". You could mention in the Conclusions section that the experiments are simulations.

A) Line 422: maultipaths.

Reviewer 2 Report

In this manuscript, the authors propose a framework useful to deal with network traffic availability. Such a tool, based on an optimization strategy, also provides a load balancing feature. The authors start from a classic graph network modeling by providing the definition of fault (and then of the availability) directly connected to the concept of link meant as an entity allowing two nodes to communicate each other. The logic behind the construction of an optimal multi-path to route traffic from multiple source nodes to a specific destination node relies on Mixed Integer Linear Programming (MILP). Then, an algorithm to split the single multi-path has been conceived. Finally, the proposed framework has been evaluated through Matlab simulations with small-sized networks with 34 nodes and 82 edges.

The paper seems to be nicely written and quite robust from a methodological point of view.

I have only a major concern. In the Related Work section (and, in general, across the whole paper), the authors seem to neglect the problem of “node availability” by only focusing on the problem of route availability. It could be interesting to investigate a bit also on the problem of a node availability. Even if a link is perfectly working, in fact, the unavailability of a node still implies a non-reachability of a network resource. Such an aspect is critical within modern network architectures (e.g., Network Function Virtualization, Service Function Chains, etc.) where, typically, the resilience is dealt by some redundancy strategies where trade-off between deployed nodes and their costs is take into account. In this connection, I suggest to mention (and, If possible to investigate the possible interconnections with the proposed framework) the following recent and credited works: "Impact of Service Function Aging on the Dependability for MEC Service Function Chain," in IEEE Transactions on Dependable and Secure Computing (2022), "Comparative Performability Assessment of SFCs: The Case of Containerized IP Multimedia Subsystem," in IEEE Transactions on Network and Service Management (2021), "Architecture-Based Reliability-Sensitive Criticality Measure for Fault-Tolerance Cloud Applications," in IEEE Transactions on Parallel and Distributed Systems (2019).

Round 2

Reviewer 2 Report

In this revised version, the Authors have addressed my main comment focused on the "node availability" problem. In the previous version, in fact, the availability problem seemed to be just related to the route. In this revised version the authors have clarified the connection between route availability and nodes/links availability, by also improving the related work section. 

In my opinion, the paper is now ready to be published in its current form.